# The Role of Preclinical Models in Creatine Transporter Deficiency: Neurobiological Mechanisms, Biomarkers and Therapeutic Development

**DOI:** 10.3390/genes12081123

**Published:** 2021-07-24

**Authors:** Elsa Ghirardini, Francesco Calugi, Giulia Sagona, Federica Di Vetta, Martina Palma, Roberta Battini, Giovanni Cioni, Tommaso Pizzorusso, Laura Baroncelli

**Affiliations:** 1Department of Developmental Neuroscience, IRCCS Stella Maris Foundation, I-56128 Pisa, Italy; elsa.ghirardini@in.cnr.it (E.G.); giuliasagona91@gmail.com (G.S.); roberta.battini@fsm.unipi.it (R.B.); giovanni.cioni@fsm.unipi.it (G.C.); 2Institute of Neuroscience, National Research Council (CNR), I-56124 Pisa, Italy; francesco.calugi@gmail.com (F.C.); f.divetta@studenti.unipi.it (F.D.V.); martina.palma@stud.unifi.it (M.P.); tommaso@in.cnr.it (T.P.); 3Department of Neuroscience, Psychology, Drug Research and Child Health NEUROFARBA, University of Florence, I-50135 Florence, Italy; 4Department of Biology, University of Pisa, I-56126 Pisa, Italy; 5Department of Clinical and Experimental Medicine, University of Pisa, I-56126 Pisa, Italy

**Keywords:** metabolism, metabolic disorders, intellectual disability, autism, epilepsy, creatine, creatine transporter deficiency, animal models

## Abstract

Creatine (Cr) Transporter Deficiency (CTD) is an X-linked metabolic disorder, mostly caused by missense mutations in the *SLC6A8* gene and presenting with intellectual disability, autistic behavior, and epilepsy. There is no effective treatment for CTD and patients need lifelong assistance. Thus, the research of novel intervention strategies is a major scientific challenge. Animal models are an excellent tool to dissect the disease pathogenetic mechanisms and drive the preclinical development of therapeutics. This review illustrates the current knowledge about Cr metabolism and CTD clinical aspects, with a focus on mainstay diagnostic and therapeutic options. Then, we discuss the rodent models of CTD characterized in the last decade, comparing the phenotypes expressed within clinically relevant domains and the timeline of symptom development. This analysis highlights that animals with the ubiquitous deletion/mutation of *SLC6A8* genes well recapitulate the early onset and the complex pathological phenotype of the human condition. Thus, they should represent the preferred model for preclinical efficacy studies. On the other hand, brain- and cell-specific conditional mutants are ideal for understanding the basis of CTD at a cellular and molecular level. Finally, we explain how CTD models might provide novel insight about the pathogenesis of other disorders, including cancer.

## 1. Introduction

Creatine (Cr) is commonly known as a nutritional supplement increasing muscle mass and performance [1]. However, Cr function is not limited to muscle fibers, as this molecule is a crucial hub for energy metabolism in every cell of the body. Since the slow diffusion of ATP and ADP is insufficient to preserve cellular ATP homeostasis, the Cr/phosphoCr (PCr) system evolved as a temporal and spatial energy buffer [2]. Cr kinase (CK) catalyzes a reversible reaction, converting Cr and ATP into PCr and ADP. PCr is a “metabolically inert” phosphagen serving as an energy reserve for fast regeneration of ATP and as a transport system of high-energy phosphates between the mitochondria, where ATP is generated, and the cytosolic compartments of energy consumption [1]. One important feature of this reaction is the presence of tissue- and cell-specific CK isoforms, with specific subcellular locations. Indeed, the Cr/PCr network relies on the differential compartmentalization of CK isoforms. Mitochondrial CK stores available cellular energy as PCr, maintaining a favorable ADP/ATP ratio for the activity of ATP synthetase, while cytosolic CK regenerates the cellular ATP pool. Cells with high-energy demand, including skeletal, smooth and cardiac myocytes, brain and kidney cells, express elevated levels of CK [3]. Furthermore, several studies have attributed to Cr a modulatory role in neurotransmission, since it is released in an activity-dependent manner in organotypic cultures of a rat brain [4,5] and acts as competitive agonist of GABA_A_ and NMDA receptors [6,7,8,9]. Finally, Cr is one of the main cellular osmolytes [10,11] and antioxidants [12,13].

In physiological conditions, Cr is obtained either from dietary sources, including meat, fish and dairy products [14] or from endogenous synthesis [15]. The latter is a two-step reaction catalyzed by the enzymes l-arginine:glycine amidinotransferase (AGAT) and S-adenosyl-l-methionine:*N*-guanidinoacetate methyltransferase (GAMT). AGAT converts glycine and arginine to guanidinoacetate (GAA) and ornithine. GAMT transfers a methyl group from S-adenosylmethionine (SAM) to GAA, generating Cr and S-adenosyl homocysteine [16]. This apparently simple pathway is complicated by the fact that most cells do not express both AGAT and GAMT, requiring the transport of the synthesis intermediate GAA between tissues. In mammals, it has been shown that the kidney contains high amounts of AGAT leading to the production of GAA, which is transported to the liver, where it is finally methylated to produce Cr. In contrast, the central nervous system (CNS) houses its own Cr-synthesizing machinery [16,17], but each cell population expresses different combinations of AGAT and GAMT, suggesting a network of intercellular cooperation for Cr synthesis and metabolism [18,19].

Following endogenous synthesis or nutritional supply, Cr is released in the bloodstream to nourish all the body cells. Cr is a polar hydrophilic molecule intrinsically unable to cross the lipid membrane and requires a specific Na^+^/Cl^−^-dependent transporter (Cr transporter, CRT) for the cellular uptake [20,21]. CRT is encoded by the *SLC6A8*gene, which is located on the long arm of the X chromosome and has a coding sequence of 13 exons [21,22]. The protein consists of 635 amino acids with a molecular mass of about 70.5 kDa [23]. This transporter is a member of a superfamily of proteins (called SLC6), which includes the transporters for the uptake of some neurotransmitters (e.g., dopamine, GABA, serotonin) and amino acids (e.g., glycine) [24]. The SLC6 family displays a common three-dimensional structure, with 12 transmembrane (TM) domains, an extracellular loop between TM3 and TM4 with N-glycosylation sites, and N- and C-termini facing the cytoplasmic side of the membrane [25,26]. CRT is tightly regulated by extracellular Cr levels, with high Cr reducing the uptake activity [16]. Moreover, cellular energy depletion inhibits CRT via the AMPK-mTOR pathway [27]. CRT is broadly expressed in different tissues [21,22,25], including the brain, where it has been predominantly detected in cortical and subcortical regions involved in motor and sensory processing, learning and memory, and in the control of affective behavior [21,28,29]. At the cellular level, CRT is expressed in oligodendrocytes and neurons, with remarkably high levels in fast-spiking parvalbumin inhibitory neurons [30,31]. It is also present in capillary endothelial cells composing the blood-brain barrier (BBB), whereas it has been detected only in a lesser amount in astrocytes [19]. Thus, Cr can enter the brain crossing the BBB, but the blood-brain transport of Cr appears relatively inefficient, at least at a mature age [32,33,34].

Inborn errors of Cr metabolism refer to three different syndromes caused by mutations in the genes encoding for AGAT, GAMT and CRT, and mainly characterized by cerebral Cr deficiency [25,35,36]. These genetic disorders share a common clinical picture with developmental delay/regression, intellectual disability (ID), severe disturbance of expressive and cognitive speech, autistic-like features, behavioral and motor abnormalities, and seizures [36,37]. While AGAT [38,39] and GAMT deficiency (AGAT-D and GAMT-D, respectively) [40] are autosomal recessive conditions affecting Cr synthesis, CRT deficiency (CTD) is an X-linked disorder impairing the cellular uptake of Cr [41,42]. Cr supplementation leads to the attenuation of symptoms in AGAT-D and GAMT-D patients. More specifically, Cr monohydrate improves epilepsy and movement disorders in GAMT-D children, with smaller beneficial effects on ID [43,44]. By contrast, AGAT-D has a more benign phenotype and patients respond better to Cr treatment [45,46,47]. Unfortunately, the same is not true for patients with CTD and this pathology is still an orphan of effective therapy. 

This review will focus on CTD, which was described for the first time in 2001 by the group of Cornelis Jakobs and Gajja Salomons [41,42]. After a brief overview of clinical aspects, diagnostic strategies and therapeutic options currently available for patients, we will describe the animal models developed for this condition in the last decade, comparing the phenotypes expressed by the different mouse and rat strains in a translational perspective. We will outline the state-of-the-art preclinical studies for this disorder, illustrating the beneficial effects and limitations of the potential therapeutic strategies tested, and offer our point of view about future directions. Finally, we will depict how the study of CTD models might provide novel insight about the pathogenesis of other disorders, including cancer.

## 2. Creatine Transporter Deficiency: Clinical Aspects and Therapeutic Management

### 2.1. Inheritance Pattern and Clinical Signs of CTD

The disruption of the *SLC6A8* gene leads to the loss-of-function of the CRT protein and the clinical manifestation of CTD (OMIM #300352). The most common genetic alterations associated with CTD are missense mutations and one-amino acid deletions, but frameshift, nonsense, translation initiation site mutations, aberrant splicing and multi-exon deletions have been detected as well. About 30% of patients carry de novo mutations, but somatic mosaicism is present in several mothers. It has been suggested that missense variants are correlated to residual CRT activity and milder phenotypes [48], but one patient with a single nucleotide change displayed severe refractory epilepsy [49]. 

The main hallmarks of CTD include global developmental delay, ID, language and speech disturbances, autistic-like traits, behavioral abnormalities, seizures and movement disorders [25,48]. CTD was for the first time described in a 6-year-old boy with a history of mild mental retardation, central hypotonia, delayed speech and language development, one episode of status epilepticus, multifocal electroencephalographic alterations, and hyperintense signal in the right posterior periventricular white matter on the magnetic resonance imaging (MRI) [41,42]. Proton-magnetic resonance spectroscopy (MRS) revealed the absence of Cr/PCr signal in the brain. Moreover, Cr levels were increased in plasma and urine, with a normal profile for other metabolites [42]. Since dietary supplementation with Cr did not improve clinical conditions or MRS signal [42], a deficit in Cr uptake was hypothesized. Consistently, sequencing of the *SLC6A8*gene revealed a nonsense mutation, resulting in an unstable and/or misfolded CRT protein that was inactive [41]. Twenty years after this first portrait, several case reports providing a detailed description of neurological and neuropsychological symptoms of CTD have been published [48,50,51,52,53,54,55,56]. Moreover, a collaborative study analyzed the phenotype of 101 patients worldwide, drawing an overview of the clinical spectrum of CTD and correlating phenotypic traits with biochemical, genetic and molecular data [48]. The authors remarked that the degree of ID in the CTD population varied from mild to severe, with intelligence quotient (IQ) scores usually in the range between 20 and 69. Cognitive functions tended to deteriorate with age and adult patients showed a more severe phenotype [48]. They underlined the possibility that the clinical presentation of CTD might include an age-dependent progression of neurological and psychiatric problems [48,57]. One of the most invalidating features of CTD is the alteration of speech and language development. The inability to speak is fairly profound, with very poor communication often restricted to a handful of words and vocalizations in most patients. Language comprehension is typically limited as well [58]. Besides the cognitive and linguistic problems, common clinical traits of CTD include attention deficit, hyperactivity, autistic-like features with social anxiety and stereotypic behavior, impulsiveness, aggressive and self-injurious behavior, and obsessive-compulsive traits [48]. Seizures were reported in about 60% of patients. The most frequent types are generalized tonic-clonic seizure and simple or complex partial seizures with or without secondary generalization, but absence of myoclonic seizures was also observed in a few patients. Children often display febrile seizures. In most cases, however, seizures are sporadic and well-controlled with antiepileptic medications [48,53,59]. Finally, other neurological symptoms, including mild to moderate motor dysfunction, hypotonia, spasticity, coordination dysfunctions, dystonia or athetosis, wide-based gait and ataxia, were consistently described [48,53]. At the structural level, brain MRI revealed abnormalities in about 70% of children, such as mild delays in myelination, T2-hyperintensities, thinning of the corpus callosum, enlarged ventricles/extracerebral spaces and cerebral/cerebellar atrophy [48]. 

This portrait clearly denotes that CTD mainly affects brain development and function. However, non-neurological symptoms, such as gastrointestinal problems (feeding difficulties, vomiting, constipation), bladder dysfunction, cardiomyopathy, and ophthalmologic abnormalities, are present in patients as well [25,48]. Poorly developed muscular mass and muscle weakness have only rarely been described [60], and Cr concentration in the skeletal muscle is apparently in the normal range [61,62]. In contrast, anomalies in electrocardiogram and echocardiographic parameters, potentially causing a severe cardiac dysfunction, have been recently reported [63].

Being an X-linked condition, CTD mainly affects males, but heterozygous females can be symptomatic as well [64]. Learning issues and mild ID are present in heterozygous female family members [64], and three index heterozygous girls have been clinically identified as patients showing ID, behavioral problems and epilepsy [65,66]. The variability of the heterozygous phenotype is mainly due to the random inactivation of the X chromosome in women. 

### 2.2. Diagnostic Methods and Prevalence

The diagnosis of CTD involves several steps. Starting from physical examination and neuropsychological profiling, the diagnostic mainstay is based on urine analysis, brain MRS, and genetic sequencing [58]. The biochemical analysis of urine quantifies the Cr:creatinine (Crn) ratio, where Crn is the breakdown product of Cr metabolism [1,25]. Previous reports established that a Cr:Crn ratio higher than 1.5 suggests a diagnosis of CTD in males [67,68]. The urinary Cr:Crn analysis is becoming a more widely available test for the metabolic screening of IDs, but metabolic alterations can be subtle in Cr disorders, often leading to inaccurate/false results [25,51,69]. MRS imaging is the gold standard for diagnosing CTD and other Cr syndromes [58,70]. However, the high costs of MRS and the need for sedation in patients with ID and behavioral disorders make this technique not accessible as a first level-screening test [51,71]. Eventually, sequencing of the *SLC6A8* gene and the assessment of Cr uptake in cultured patient fibroblasts are required to conclusively validate a diagnosis of CTD [52,58].

Despite a precocious onset of symptoms, the average age of diagnosis of CTD ranges from 2 to 20 years [72]. The identification of early clinical indicators, alerting pediatricians to seek a specialty referral and/or conduct a urine screen, might accelerate the diagnosis. In a recent retrospective study of parent reports and medical records, the authors observed that early feeding or weight gain problems presenting together with delays in motor function milestones should encourage primary care providers to consider rare metabolic disorders [72]. Further research will examine the natural history of CTD, providing novel tools for the follow-up of this condition and possibly monitoring the efficacy of potential therapies.

Overall, CTD emerges as a relatively common cause of X-linked ID. According to several reports, the worldwide prevalence of CTD could be estimated between 0.25 and 1% in males with ID [51,73,74], while 0.024% of females might be heterozygous carriers [75]. The prevalence increases to 4.4% when focusing on mental retardation of unknown origin [76]. CTD patients are often misdiagnosed with cerebral palsy in early postnatal age and with Autism Spectrum Disorder (ASD) in later development. A recent study presented the case of a 6-year-old boy showing speech delay, lack of several social milestones, including shared attention, imaginative play, name responding and pointing, stereotypical movements, attention deficit, hyperactivity, avoidant and restrictive eating habits, and excessive fluid intake. One episode of generalized tonic-clonic seizure was reported at 3 years of age. This clinical picture initially led to the diagnosis of ASD according to the Diagnostic and Statistical Manual of Mental Disorders, 5th edition (DSM-5) criteria, with accompanying language impairment and ID. MRS, however, detected a significantly decreased Cr peak, directing to diagnostic investigations towards disorders of Cr metabolism. Molecular analysis, indeed, revealed a novel, frameshift variant of the *SLC6A8* gene, confirming a diagnosis of CTD [77]. This paradigmatic case report underlines the significance of extending newborn screening for metabolic disorders to Cr deficiency syndromes.

### 2.3. Therapeutic Perspectives in CTD Patients

Contrary to other Cr deficiency syndromes [78,79,80], efforts to rescue Cr levels in the CTD brain by dietary supplementation had limited success [20]. Cr supplementation, alone or combined with arginine, glycine, and/or S-adenosylmethionine, failed to restore brain Cr content and to ameliorate clinical signs in the majority of patients with CTD [81,82,83,84,85]. Indeed, dietary strategies increased cerebral Cr and improved clinical parameters only in milder cases with residual brain Cr, and possible residual CRT protein function [86,87,88]. Moreover, long-term administration of arginine and glycine might induce adverse effects [81,82,83,84].

A recent work suggested that betaine supplementation exerts positive effects on the CTD phenotype and is well tolerated [83]. However, this is an open trial without a placebo control and it was limited to the observation of two patients; further studies in a larger number of subjects following a well-designed study protocol are required to establish the true therapeutic value of betaine. 

Altogether, these data show that no satisfactory treatments are available yet for CTD. The mainstay of care is support for families, envisaging palliative therapeutic strategies for managing seizures and behavioral problems. 

## 3. Modeling CTD Features in Animals

Although rare, CTD is a major social issue, with patients demanding lifelong assistance, and the severity of the disease burdening families and the health-care system. Therefore, there is a pressing need to identify effective therapies for this disorder. Moreover, little is known about the causal chain of events leading from Cr depletion to the impairment of multiple key behavioral domains. Delineating how the metabolic failure in CTD might converge into brain function alterations is a prerequisite of paramount importance for the development and assessment of therapeutic strategies. 

Animal models are foundational resources for studying basic biological processes, disease pathogenesis and novel therapies. Over the past ten years, several models of CTD have become available. They can be grouped into two main categories: whole-body and brain- (or cell-) specific lines. In the next sections, we will provide an overview of the different approaches used to generate CTD models, comparing cognitive, behavioral and functional phenotypes expressed by these animals.

### 3.1. Whole-Body Rodent Models 

Four whole-body CTD models have been developed so far. Two of them are knockout (KO) mice carrying the deletion of exons 2–4 (KO(2-4)) [89] or 5–7 (KO(5-7)) [90] in the *SLC6A8* gene, resulting in the absence of a functional CRT protein. With a similar method, a third KO mouse has been obtained by targeted disruption of the gene through the insertion of a β-galactosidase/LacZ reporter expression cassette between the exons 5 and 7 [91]. More recently, Duran-Trio and colleagues used CRISPR/Cas9 technology to generate a rat knock-in (KI) strain carrying the Y389C mutation of *SLC6A8*, a tyrosine-to-cysteine missense variation detected in CTD patients [92], which results in the complete suppression of the transporter activity [48,93]. As a model for CTD, the KI line has the advantage of carrying one of the mutations responsible for the disease in the appropriate protein context, potentially providing a more faithful picture of the pathological situation.

The comparative review of published studies revealed that the most relevant features of CTD are consistently present across these animal models. First, brain Cr levels were invariably depleted, although a mismatch among measured concentrations stands out. The discrepancy in absolute values, ranging from the total absence of Cr [89] to a reduction by 70–90% with respect to wild-type (WT) littermates [90,93,94,95], however, is likely attributable to the use of detection methods with different sensitivities, such as colorimetric assays, gas or liquid chromatography coupled to mass spectrometry, or reversed phase HPLC and MRS. Cr decrease was also reported in several peripheral organs, including skeletal muscle, heart, kidney, testes and serum, while high Cr was found in urine, as expected [89,90,91,93,94,96] (Figure 1). It should be noted that residual Cr was present in most tissues. This may result from compensatory uptake via alternative transporters. Indeed, the paralogous gene *SLC6A10* is present in the testis and brain [22,97], the second putative creatine transporter encoded by *SLC16A12* gene is highly expressed in the kidney and retina with low levels in the brain [98], while Loike and colleagues suggested that a fraction of Cr transport into cultured rat L6 muscle cells could take place via a sodium independent mechanism [99]. Additionally, endogenous synthesis may contribute to maintaining Cr levels, and a moderate rise in GAA was observed in various body districts of mutant animals, suggesting that Cr depletion might upregulate AGAT activity [90,93,100]. Since these compensatory mechanisms may represent possible therapeutic targets, further studies will be needed to elucidate how tissues can partially preserve Cr levels despite the loss-of-function of CRT.

Another distinctive general trait shared by all the whole-body models was reduced weight [89,90,91,93,94,95,96]. Although an alteration of physical features is not obvious in most CTD patients, low weight gain, slender build, decreased height and poor developed muscular mass have been described [25,72], corroborating the translational value of experimental animal strains. 

As mentioned above, the hallmark manifestation of CTD is ID. This aspect can be effectively assessed in rodents through a series of tests evaluating different types of memory and cognitive performance [101]. Using the Y maze, the object-recognition test (ORT), the Morris water maze (MWM), and the contextual and cued fear conditioning (FC), researchers demonstrated that human ID is well recapitulated by CTD models, as they exhibited deficits in most of these tasks. More specifically, both KO(2-4) and KO(5-7) mice showed an altered performance in ORT and MWM, revealing the impairment of declarative and spatial memory [89,90,94]. In contrast, ORT was not affected in KI rats [93]. This discrepancy might depend on the different impact of CRT loss in the behavioral physiology of the two species or on variations in the experimental settings used for testing. In KO(2-4) mice, compromised associative memory was observed as well, with reduced freezing responses to cued and contextual fear [89]. Mild deficits in the MWM were also described in heterozygous KO(2-4) female mice, which otherwise showed no defects in ORT and FC [102]. On the other hand, working memory in the Y maze was damaged in both mutant mice and rats [90,93,94]. Interestingly, the Y maze test was found to be the single variable most informative for the discrimination between KO and WT animals [103]. Ease of testing is a crucial factor in the assessment of complex phenotypes, and the availability of a simple and reliable tool for CTD classification is highly valuable as potential first-line screening in preclinical studies of therapeutics (Figure 1). 

Since numerous non-cognitive factors may lead to impaired performance in memory tasks, it is worth noting that most studies reported a lack of anxiety and stress-related behaviors in CTD animals [89,90,104], nor were there changes in the affective behavioral domain. Despite a shorter latency to immobility in the tail suspension test, KO(2-4) mice appeared to be rather resilient to induced depression [104]. In contrast, the analysis of locomotor behavior in KO(2-4) mice revealed the presence of hyperactivity, while KO(5-7) animals were found to be less active at least during the dark phase [89,90]. These conflicting results might be explained by the different experimental conditions of testing, as locomotor activity of KO(2-4) mice was assessed for one hour in a dedicated chamber, whereas KO(5-7) animals performed the test over 24 h in the home-cage. The presence of a new, challenging environment as opposed to a familiar one could account for the observed differences. Accordingly, behavioral outputs in patients suffering from Attention-Deficit/Hyperactivity Disorder (ADHD) tend to depend on different contexts or situations [105,106]. Altogether, these observations suggest that a slight, novelty-induced hyperactivity might characterize the phenotype of KO mice (Figure 1). 

In the last 5 years, growing evidence has highlighted the relevance of Cr in establishing and maintaining proper cognition throughout different developmental phases of postnatal life. Indeed, the time-dependent deletion of the *SLC6A8* using a drug-inducible Cre recombinase led to cognitive defects only when tamoxifen treatment was started in newborns (postnatal day 5, PND5), whereas no deficits emerged in animals with late deletion of the gene [95]. Accordingly, it has been shown that Cr biosynthesis and uptake are essential during embryonic development [107], with the embryonic CNS predominantly depending on external Cr supply. These findings indicate that CRT deficiency might induce permanent damages as early as in utero [108]. On the other hand, a longitudinal analysis of KO(5-7) mice unveiled the effects of long-term Cr shortage, demonstrating that cognitive impairment progressively deteriorates throughout adulthood in these animals and is accompanied by signs of early brain senescence, including synaptic loss, microglial activation, reduced hippocampal neurogenesis and lipofuscin deposition [94]. Moreover, the lifespan of KO(2-4) mice was significantly shorter with respect to WT littermates [96].

Since ASD traits are often present in the clinical picture of CTD patients, CTD models were also examined for sociability and stereotyped movements [109,110,111]. Of the two ASD domains, only stereotypies were straightforward in CTD models. An increase in the time spent grooming themselves was in fact reported in both KO(5-7) mice and KI rats [93,94]. Using rotarod motor learning as a proxy for acquired repetitive behaviors in mice, it has been shown that the deletion of exons 5–7 enhanced the formation of repetitive motor routines [94]. Conversely, no defects in social interactions were observed, as KO(2-4) mice, KO(5-7) mice and KI rats displayed normal social preference and social novelty [89,93,94] (Figure 1). 

Finally, one of the most relevant pathological features of CTD is epilepsy, which occurs in approximately 60% of patients [25]. To date, this aspect was investigated in a single study, which reported the presence of spontaneous seizures and increased susceptibility to kainate-induced epilepsy in KO(5-7) mice [103] (Figure 1). Neurodevelopmental disorders often include epilepsy in their clinical portrait, but the concordance of the epileptic phenotype between patients and their model counterparts has rarely been reported [112]. These data therefore highlight the good face validity of CTD models. Using electroencephalogram (EEG) recordings and intrinsic optical signal imaging, the same study identified a set of quantitative biomarkers for monitoring brain function in CTD. Specific alterations in theta, alpha, beta and gamma EEG frequency bands and a paradox increase in hemodynamic cortical response were found in KO(5-7) mice. The clinical relevance of EEG spectral changes was demonstrated by the comparable dysfunction in theta, alpha and, to a lesser extent, gamma bands in CTD patients [103]. Moreover, the growing accessibility of non-invasive techniques, such as functional near-infrared spectroscopy (fNIRS), will allow the study of cerebral blood flow in these individuals [113]. Although reliable means for the diagnosis of CTD are already available, the biomarkers discovered in this work will be important, as they will allow the standardization of the follow-up of patients and represent additional, informative tools to investigate the efficacy of therapeutic strategies in preclinical, and hopefully clinical, studies.

Animal models offer the unique chance to study the pathological determinants of CTD to a cellular and molecular level. Despite no evidence of gross alterations in brain weight and anatomy [89,94,96], a more subtle reorganization of neural circuits, including altered levels of neurotransmitters like glutamate, GABA and serotonin [89,93,104], reduced density of inhibitory synapses, dysregulated microglia activation, and impaired hippocampal neurogenesis [94], suggesting that CTD pathogenesis might present a multicellular profile with a complex network of cell-autonomous and non-autonomous effects. A relatively unexplored role of Cr is that of regulation of the immune system, where it contributes to the fine tuning of both the innate and the adaptive response [114]. Even though a detailed discussion of this matter goes beyond the scope of this review, animal models of CTD played a pivotal role in understanding some of the Cr mechanisms of action in this picture. For example, Ji et al. showed that Cr was essential to regulate macrophage polarization towards an anti-inflammatory profile. CRT loss-of-function caused a pro-inflammatory reprogramming of these cells, boosting the host defense upon bacterial infection in CRT KO mice [115]. Although enhanced inflammatory responses may have positive outcomes in some cases, there is also the other side of the coin. As mentioned above, KO(5-7) mice exhibited a marked increase in activated microglia [94], the resident macrophage population of the brain, and it is well-known that excessive microglial inflammation is implicated in many neurological diseases [116]. It is still not clear to what extent microglia express the *SLC6A8* gene [31,117] or if polarization in these cells is governed by the same factors operating in their peripheral counterparts [118,119], but the study from Ji and colleagues opens up the possibility that neuroinflammation in CTD models might be due to a similar mechanism. 

Furthermore, Cr deficiency profoundly disrupts body metabolism [94,120], resulting in a compensatory upregulation of proteins involved in energy homeostasis and mitochondrial activity, and in a significant increase in oxidative stress [91,94,120,121]. Although further studies will be needed to better understand the molecular pathophysiology of CTD, we believe that antioxidant drugs and inflammation inhibitors might be valid therapeutic venues for CTD.

Besides the neurological phenotype, the main alterations exhibited by whole-body CRT KO models were found in the skeletal muscle, where reduced Cr [91,94,100,122] was associated with muscle fiber atrophy, vacuolar myopathy [91,96], and significant abnormalities in strength, endurance, motor coordination and learning [89,91,94]. Despite a normal heart histological structure [96], ECG and echocardiographic anomalies were described [63]. The deficits in the structure and function of the skeletal muscles are in apparent contrast with the normal levels of Cr detected in the same tissue of two CTD patients [61,62]. However, hypotonia is the second most common comorbidity in CTD patients [25,86] and a more thorough examination of Cr levels in the muscle of patients will be needed before dismissing the muscle phenotype of CRT KO mice as inconsistent with clinical data [120]. 

The open debate concerning the extent of the muscular phenotype in CTD patients and the validity of the murine models in recapitulating this aspect, combined with the finding that extraneural factors might underlie behavioral deficits in other neurodevelopmental disorders [123,124], has led the CTD scientific community to investigate the phenotype of brain-specific animal models.

### 3.2. Brain- and Cell-Specific Conditional Mouse Models

Two different brain-specific KO mouse models of CTD have been generated in the last few years [94,125,126]. Using the Nestin promoter [127] to drive Cre-recombinase expression, Baroncelli et al. removed exons 5–7 (Nes-KO(5-7)) [94,125] and Udobi et al. exons 2–4 (Nes-KO(2-4)) from the *SLC6A8* gene in neurons, glia and endothelial cells forming the BBB [125]. 

Brain-specific CRT loss-of-function was confirmed by a significant decrease of Cr content in various brain regions and a parallel rise in the local synthesis of GAA. Peripheral tissues, including muscle, the heart and the kidney, were not affected [94,125] (Figure 2). Regular age-dependent body growth and normal forelimb force were also observed in these animals as a result of preserved Cr levels in the muscle. Only in the adult age (i.e., PND100), Nes-KO(5-7) mice started to show a slight decrease in body weight with respect to control littermates, suggesting a possible disturbance in the neural mechanisms regulating food intake and whole-body metabolism [125] (Figure 2). Even though weight reduction was relatively modest, future studies will be needed to investigate hypothalamic function in Nes-KO mice, and to assess the possible correlation between the lower body mass and the hyperactivity recorded in these animals [125,126]. 

There is general consensus that Nes-KO mice have a global impairment in cognitive functions ranging from procedural skills to declarative and spatial memory, as demonstrated by their altered performance in the Y maze, ORT, MWM, and contextual and cued FC [125,126] (Figure 2). In line with the results obtained in the whole-body strain, cognitive failure reflected an impairment of hippocampal neurogenesis in PND180 Nes-KO(5-7) mice, showing a lower density of Ki67-positive cells in the dentate gyrus [94]. The behavioral data, however, were not entirely consistent between the two studies. Nes-KO(2-4) animals showed deficits in the ORT and MWM already at PND60 [126], whereas Nes-KO(5-7) mice manifested a late-onset (PND100) memory disruption accompanied by an age-dependent increase of cognitive frailty [125]. This discrepancy could be explained by differences in the experimental setting, but also in the motivational drive of the animals: indeed, Udobi et al. reported that Nes-KO(2-4) mutants also fail in the cued version of the MWM, which is specifically designed to test the mouse motivation to reach a visible platform. 

Interestingly, alterations of cognitive performance in Nes-KO mice were not due to motor dysfunction or high levels of anxiety: motor skills and muscle strength were totally preserved in these animals, and the behavior measured in the open field arena was in the range of normal values. Although locomotor activity was slightly increased in Nes-KO animals, it is unlikely that moderate hyperactivity could account for the observed cognitive dysfunction [102,125,126]. On the other hand, hyperactivity and attention deficits are commonly described in CTD patients [25,128]. Since hyperactivity in ubiquitous KO mice is presumably masked by the motor problems due to peripheral Cr deficiency, the presence of this phenotype in conditional mice represents an original finding, making them a viable model to study certain aspects of this disorder. 

In contrast to the whole-body mutants, the endophenotype of Nes-KO animals did not include an increment of stereotyped movements and repetitive behaviors, suggesting that extraneural Cr might play a central role in the onset of autistic-like traits [125] (Figure 2). Studies of the incidence of spontaneous epilepsy and susceptibility to proconvulsant treatment are not yet available for conditional models.

Altogether, these studies demonstrate that brain Cr deficiency leads to significant learning and memory deficits, providing two valuable models for investigating nervous-system specific mechanisms of CTD. 

In search of the cellular basis of CTD pathophysiology, it was demonstrated that deleting *SLC6A8* from forebrain excitatory neurons reproduced at least partially the neurological phenotype of CTD. The decrease of brain Cr content in CaMKII-KO(2-4) mice was mirrored by a broad-spectrum disruption of cognitive abilities in the adult animals with no alteration of motor functions [129]. Moreover, home-cage activity increased in CaMKII-KO(2-4) mutants, suggesting that glutamatergic neurotransmission is crucially involved in the expression of ADHD-like traits characterizing CTD (Figure 2).

In contrast, the selective loss of *SLC6A8* in dopaminergic neurons (DAT-KO(2-4) mice) led to spontaneous hyperactivity, while sparing general motor and cognitive functions [130] (Figure 2). These data are consistent with the major involvement of the dopaminergic system in ADHD [131,132].

According to the previous literature about other neurodevelopmental disorders [133,134], these results highlight that choosing the proper animal model is of paramount importance for the preclinical drug development pipeline. Although the conditional CRT KO mice did develop cognitive impairment in their adult age, they failed to recapitulate the early pathological phenotype and the autistic-like traits of CTD patients, indicating that they are a suboptimal tool for the assessment of potential therapeutics of translational value. Thus, whole-body mutants emerge as the preferable model of CTD. However, brain- and cell-specific mice might be meaningful for in-depth investigations of the pathogenetic mechanisms underlying the neurological deficits induced by Cr deficiency and the role of Cr in the maturation of different cell populations of the nervous system.

### 3.3. Preclinical Testing of Potential Therapeutic Strategies

Studies on Cr deficiencies due to mutations of biosynthetic enzymes have shown that the replenishment of brain Cr is effective in ameliorating the clinical manifestation of the disease [78,79,80], proving the reversibility of the pathological effects caused by Cr deficit. Thus, *in vitro* systems and animal models have been extensively employed for preclinical assessment of different strategies aiming to reinstate brain Cr levels in CTD patients.

Cyclocreatine (cCr) is an artificial analog of Cr, which can enter cells independently of CRT [135,136] and can be phospho/dephosphorylated by CK [136,137], mimicking the metabolic function of Cr. This compound has been proposed as a possible replacement of endogenous Cr. This approach has been first assessed in brain-specific CRT KO mice, resulting in improved learning and memory [129]. These findings prompted a more rigorous preclinical study exploring the therapeutic efficacy of a long-term treatment with cCr at three different doses in animals where the ubiquitous CRT loss aligns better with the genetics and early onset of CTD [138]. The minimum efficacious dose (MED) of cCr for improving both the cognitive and the epileptic phenotype was 46 mg/kg, whereas 14 mg/kg was sufficient to ameliorate stereotyped behaviors [138]. No detrimental repercussions of cCr administration have been observed [138]. However, despite efficient kinetics of brain penetration, oral cCr treatment displayed only a partial efficacy in reversing the pathological CTD phenotype, with moderate effects upon cognitive functions and against seizures [138]. This is likely ascribable to the inability of cCr to release free Cr [16] and to its low affinity for CK [137,139], preventing cCr from restoring all natural Cr functions. In addition, high doses of cCr have been recently reported to increase seizure incidence and brain vacuolation in rats, possibly because of the accumulation of toxic metabolites [140,141]. Thus, the window between the “No Observed Adverse Effect Level” (NOAEL) and MED was narrower than the safety margins established by the US Food and Drug Administration (FDA), and the development of this drug has been discontinued.

A parallel research line focused on creatine esters as a possible tool for restoring cerebral Cr content. Intriguingly, these Cr derivatives are highly lipophilic and can passively diffuse inside the cells [142], where they are processed by cellular esterases delivering free Cr [143]. Creatine benzyl ester (CBE) has been shown to increase the intracellular Cr and PCR pool in mouse hippocampal slices. However, CBE was rapidly degraded to benzyl alcohol and Crn with possible toxic outcomes for brain cells [142]. Creatine ethyl ester (CEE) appeared to enter fibroblasts obtained from CTD patients [144], but further studies revealed that the lipophilicity of this compound was not sufficient to cross the plasmatic membrane independently from the Cr transporter and that its half-life was very short in aqueous solution [145]. Despite the positive effects on synaptic transmission reported in WT hippocampal neurons [145], CEE has poor chemical stability and is particularly vulnerable to gastric environment and plasma esterases [146]. Accordingly, CEE administration in CTD patients did not change Cr levels and did not induce a significant improvement in neuropsychological performance [144]. More recently, a screening of Cr fatty esters on cellular models revealed that the incorporation of these prodrugs in neurons, astrocytes and endothelial cells depends on the length of the carbonyl chain. This study identified dodecyl creatine ester (DCE) as the compound with the most interesting features in terms of brain penetrability and diffusion [147,148]. DCE was reported to increase Cr levels in fibroblasts of CTD patients [147] and intranasal administration of DCE-loaded microemulsion enhanced ATP in the brain of CRT KO mice, leading to a better performance in declarative memory [149]. Another Cr ester with high lipophilic character, di-acetyl creatine ethyl ester (DAC), was shown to counteract the harmful effects of CRT pharmacological blockades in brain slices at both the biochemical and electrophysiological level [150]. 

An alternative strategy to overcome the impermeability of plasmatic membranes to hydrophilic elements is to couple Cr to aminoacids or other molecules, which have their own specific carrier [151]. Examples of such compounds are Cr-Gly-OEt and Cr-NHEt, which were capable of increasing the cellular content of Cr in brain slices after the block of CRT [152], but have never been tested in vivo. The same was true for Cr gluconate [153]. 

Finally, it has been recently suggested that some of the mutations in *SLC6A8* associated with CTD symptoms might engender folding defects in the transporter, altering the intracellular distribution and the function of the protein. The chemical chaperone 4-phenylbutyrate (4-PBA) restored surface expression as well as substrate uptake for several folding-deficient CRT mutants, providing a promising rationale for the study of 4-PBA in KI animal models [154].

In summary, these data suggest that Cr esters endowed with proper delivery systems to reach the brain parenchyma may be a good drug candidate for CTD. Despite the proof-of-concept demonstration that DCE microemulsion did improve the phenotype of KO(2-4) mice, however, further studies on CTD animal models are mandatory to determine in vivo the efficacy and the therapeutic window of these drugs, and to assess their toxicological profile. Moreover, preclinical multicenter randomized controlled trials testing the compounds in different animal models of CTD will significantly strengthen the reliability of the results obtained, setting the foundation for the translational success [155].

## 4. Role of Metabolism in Cancer Progression: Novel Insights from Creatine Models

The Cr system is currently the focus of renewed interest for the bioenergetics of cancer metastasis [156]. Cancer is a leading cause of mortality worldwide, with metastatic progression being the predominant cause of death [157]. The high prevalence of this devastating disorder and the lack of effective treatments for preventing metastases demands a greater knowledge of the underlying biological mechanisms [158]. 

Cellular energy state and metabolism emerged as key features of tumorigenesis and cancer progression [159,160,161]. By regulating the cellular reservoir of high-energy phosphates [16], the Cr/PCr system actively drives the proliferation of tumor cells and the colonization of other tissues. Accordingly, the expression of CK, AGAT and CRT was induced in a variety of tumors, increasing intracellular Cr and ATP levels [162,163,164,165]. This energetic boost facilitates cancer cell division and survival [16,163,164,166,167,168,169,170], and regulates metastasis by affecting cell motility and migration [161,164,165,170,171]. For instance, it has been shown that *SLC6A8* overexpression promotes the proliferation and invasion of human tumor cells *in vitro*, whereas the knockdown of CRT suppresses these processes [164,170]. Conversely, CK depletion reduced the survival of colon cancer cells under hypoxia, inhibiting liver metastasis [172]. The authors of this work hypothesized that migrating malignant cells release the CK protein in the extracellular milieu in order to convert free ATP and Cr into PCr that is then assimilated by cancer cells and exerts a protective effect promoting their adaptation to the colonized microenvironment. Consistently, clinical studies detected CK activity in the serum of patients with various forms of malignancies [173,174], and a higher expression of CK and CRT in metastases with respect to primary tumors [172]. In addition, Loo and colleagues found that exogenous PCr was sufficient to significantly enhance metastatization in animal models, whereas cancer cell lines depleted of the *SLC6A8* gene displayed substantially reduced metastatic activity, even when CK was overexpressed [172]. A recent work also showed that, besides dietary Cr, endogenous synthesis plays a crucial role in metastases, as the knockdown of the AGAT gene in colorectal cancer cells significantly suppressed *in vivo* the invasion of the liver tissue, prolonging the life span of mice [165].

In summary, the Cr/PCr system represents a powerful mechanism of survival for cancer cells and targeting this metabolic network might have a therapeutic potential for the treatment of this disease. In this framework, cCr, alone or in combination with common chemotherapeutic agents, has been shown to have antiproliferative activity in tumor cell lines [16,166,175,176,177,178], counteracting tumor growth and metastatic cascades in vivo [16,172,176,177]. The antitumor activity of cCr is probably related to the inhibition of the CK enzyme [16], but further studies are needed to clarify the mechanisms of action of this Cr analog. Moreover, decreased Cr and CK levels have been reported for other forms of cancer [179,180,181], suggesting that the role of the Cr/CK system is not straightforward in tumor pathogenesis and that the therapeutic value of this metabolic pathway in cancer therapy might be multifaceted [182].

Finally, Cr is critical to promote antitumor activity in the cytotoxic T CD8^+^ cells, and the response to tumor challenge is severely impaired in KO(2-4) mice [183]. These results add up to the accumulating evidence that implicates CRT in cancer, encouraging a close survey of cancer incidence in CTD patients [128].

## 5. Conclusions and Future Directions

Creatine transporter deficiency is an X-linked metabolic disorder affecting about 0.25%–1% of males with ID. CTD prevalence, however, is likely underestimated because of the frequent misdiagnosis of patients, and is expected to rise as soon as the panel of Cr deficiency syndromes will be included in the prenatal or newborn screening for genetic disorders. 

Currently, there is no cure for CTD and the mainstay of care includes pharmacological management of epilepsy and support for families. Thus, the research of therapies is a fundamental challenge. Unravelling the pathogenesis of Cr deficiency is an essential step to disclose novel therapeutic targets, but still too little is known about the molecular and cellular mechanisms affected by Cr depletion. 

Animal models are crucial tools to investigate disease biology and to devise effective therapeutics. In recent years, *SLC6A8* loss-of-function rodent models well-recapitulating the human disease have become available, and preclinical testing highlighted that pharmacological therapy with lipophilic Cr analogs might be a promising approach with high translational value. However, variations in phenotypes that have been observed in animal models highlight the importance of validating the therapeutic efficacy of a specific treatment in multiple models and laboratories. 

As a monogenic condition, CTD is, in principle, a very good candidate for gene therapy. Vector-based delivery of functional *SLC6A8* genes and CRISPR-CAS9 repair of specific mutations are currently under active investigation and might provide an alternative option for the treatment of patients. Finally, the efforts spent on dissecting dysfunctional brain circuits and altered molecular processes in CTD might pave the way for drug repurposing, an attractive option because of its low costs and short development timelines.

It is also worth stressing that Cr exerts pleiotropic functions in different cell types and alterations of this metabolic pathway have been reported to play a role in several disorders beyond Cr deficiencies, including muscular dystrophies, cardiac diseases, obesity and cancer. Thus, the study of CTD animal models might provide novel insight in Cr biology, showcasing the therapeutic potential of targeting the Cr/CK/PCr system in a wide spectrum of human pathological conditions.

## Figures and Tables

**Figure 1 genes-12-01123-f001:**
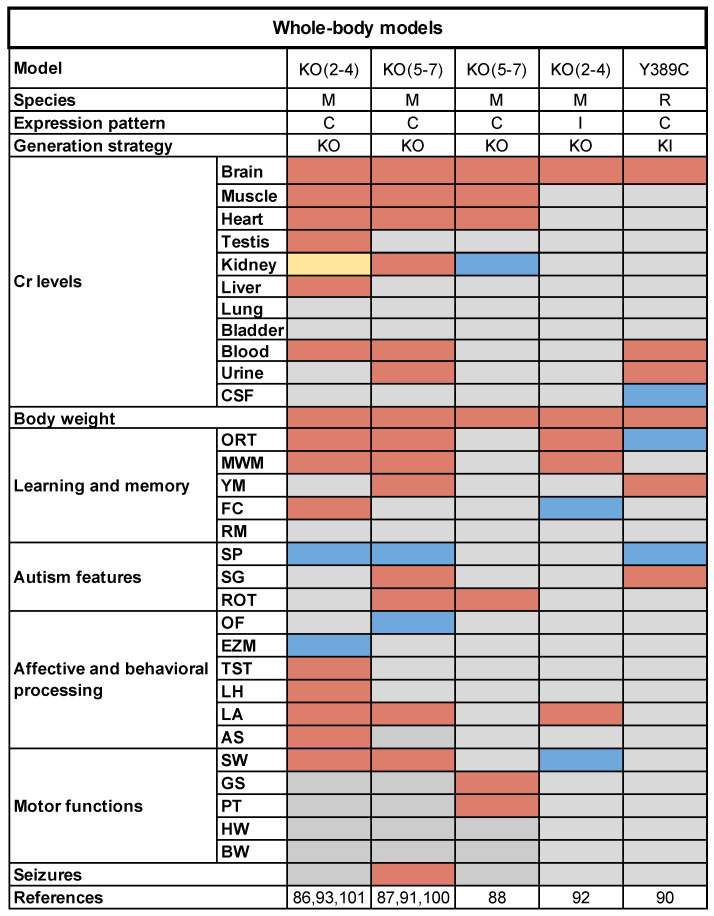
Heatmap representation of the cluster of whole-body animal models for CTD carrying different deletions/mutations of the *SLC6A8* gene. Species (M: mouse; R: rat) expression patterns of a mutated allele (C: constitutive; I: inducible) and model generation strategy (KO: knock-out; KI: knock-in) were also cited. The phenotypic traits for each model are indicated (ORT: object-recognition task; MWM: Morris water maze; YM: Y maze; FC: fear conditioning; RM: radial maze; SP: social preference; SG: self-grooming; ROT: rotarod; OF: open field; EZM: elevated zero maze; TST: tail suspension test; LH: learned helplessness; LA: locomotor activity; AS: acoustic startle; SW: swimming behavior; GS: grip strength; PT: pole test; HW: hang wiring; BW: beam walk). Red squares mean significantly higher or lower performance with respect to WT animals. Blue squares refer to phenotypes with no differences between genotypes. Yellow square indicated mixed evidence. Grey squares are for not investigated tasks.

**Figure 2 genes-12-01123-f002:**
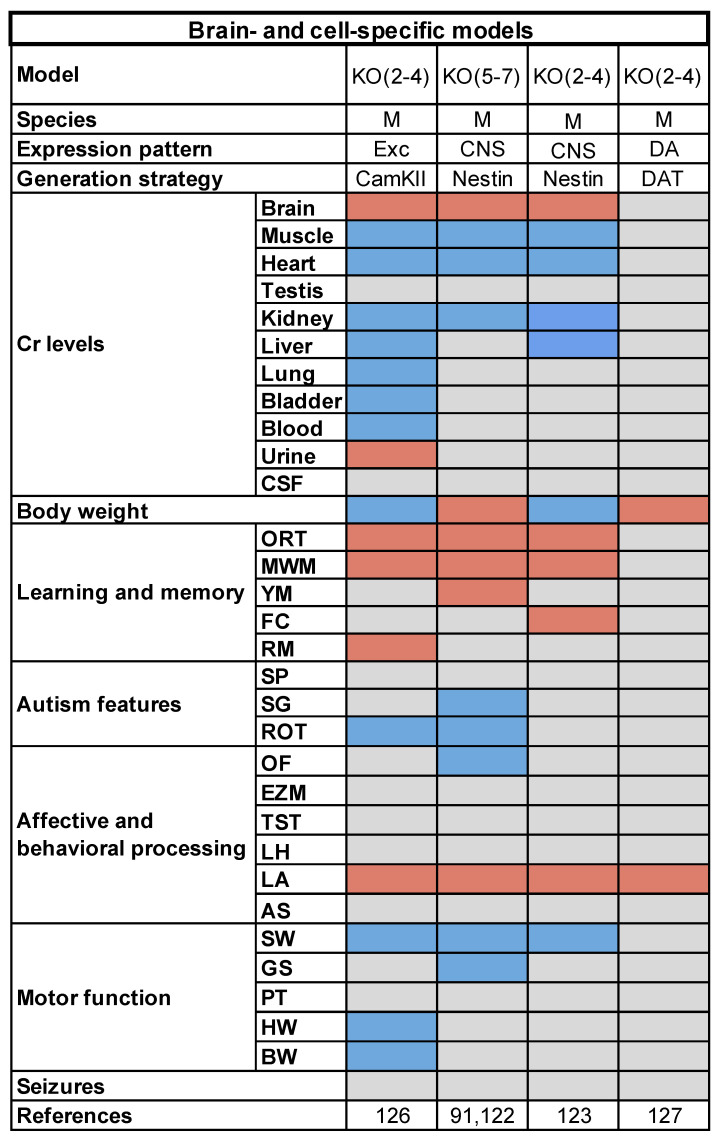
Heatmap representation of the cluster of conditional models for CTD carrying different deletions of *SLC6A8*gene restricted to specific brain districts. Species (M: mouse), expression pattern of mutated allele (Exc: excitatory neurons; CNS: postmitotic neurons, glial cells and BBB endothelial cells of central nervous system; DA: dopaminergic neurons) and model generation strategy (the promoter driving Cre-recombinase expression) were also cited. The phenotypic traits for each model are indicated (ORT: object-recognition task; MWM: Morris water maze; YM: Y maze; FC: fear conditioning; RM: radial maze; SP: social preference; SG: self-grooming; ROT: rotarod; OF: open field; EZM: elevated zero maze; TST: tail suspension test; LH: learned helplessness; LA: locomotor activity; AS: acoustic startle; SW: swimming behavior; GS: grip strength; PT: pole test; HW: hang wiring; BW: beam walk). Red squares mean significantly higher or lower performance with respect to WT animals. Blue squares refer to phenotypes with no differences between genotypes. Grey squares are for not investigated tasks.

## Data Availability

Not applicable.

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
