# Peer review of "The Role of Preclinical Models in Creatine Transporter Deficiency: Neurobiological Mechanisms, Biomarkers and Therapeutic Development"

_genes, 2021, doi:10.3390/genes12081123_

Round 1

Reviewer 1 Report

This manuscript of Ghirardini et al. provides an excellent review of the preclinical models for creatine transporter deficiency (CTD). It is comprehensive, up-to-date and it reads very well.

There are a few minor aspects, though, that I would appreciate it they could be addressed by the authors.

Firstly, the manuscript does not mention or assess the experimental setting in the pre-clinical models with respect to exogenous creatine exposure. Is there any difference in experimental outcomes when animals are housed in a creatine-free environment, i.e. fed with creatine-free mouse/rat chow, mutants separated from het and wt littermates, versus animals that received even spurious amounts of creatine? While the former may allow a better elucidation of the role of endogenous creatine synthesis in the context of CTD, the latter allows a better comparison to the situation in CTD patients that are exposed to exogenous creatine through lifetime.

Secondly, there is a cardiac phenotype in CTD that was recently discovered and published by Levin and coworkers (Levin et al 2021). This important finding should be added to this excellent review.

Lastly, the prevalence of 4% of CTD in males with non-syndromic mental disability as stated in ‘conclusion and future directions’ is likely an over-estimate. This figure is based on one single center study and it was only published as an abstract (#73, Lion-Francois et al. 2006). A prevalence of 0.25-1% in male ID, as it is mentioned on page 4 in the manuscript, is certainly more realistic.

Literature: The first synopsis of creatine deficiency syndromes has been published in 2003 (Schulze 2003). Although it is based on the first observations in the few patients with AGAT, GAMT and CTD known at the time this synopsis holds true up to the day and could be added to the comprehensive body of literature in this manuscript.

Levin MD, Bianconi S, Smith A, et al (2021) X-linked creatine transporter deficiency results in prolonged QTc and increased sudden death risk in humans and disease model. Genet Med. https://www.ncbi.nlm.nih.gov/pubmed/34050321

Schulze A (2003) Creatine deficiency syndromes. Mol Cell Biochem 244: 143-150.

Author Response

Response to Reviewer #1: (reviewer’s comments are in italics)

- This manuscript of Ghirardini et al. provides an excellent review of the preclinical models for creatine transporter deficiency (CTD). It is comprehensive, up-to-date and it reads very well. There are a few minor aspects, though, that I would appreciate it they could be addressed by the authors.

 We thank the reviewer for the positive evaluation of our work. We hope that the revised manuscript will address all the points raised by the reviewer.

- Firstly, the manuscript does not mention or assess the experimental setting in the pre-clinical models with respect to exogenous creatine exposure. Is there any difference in experimental outcomes when animals are housed in a creatine-free environment, i.e. fed with creatine-free mouse/rat chow, mutants separated from het and wt littermates, versus animals that received even spurious amounts of creatine? While the former may allow a better elucidation of the role of endogenous creatine synthesis in the context of CTD, the latter allows a better comparison to the situation in CTD patients that are exposed to exogenous creatine through lifetime.

 The reviewer raised a very relevant point. However, only two studies describing preclinical models provided some details about the food composition and the animals’ housing conditions. Thus, we are not able to reliably compare these aspects in different experimental settings.

- Secondly, there is a cardiac phenotype in CTD that was recently discovered and published by Levin and coworkers (Levin et al 2021). This important finding should be added to this excellent review.

 As suggested by the reviewer, we included the findings about the cardiac phenotype of CTD patients and mice in the review.

- Lastly, the prevalence of 4% of CTD in males with non-syndromic mental disability as stated in ‘conclusion and future directions’ is likely an over-estimate. This figure is based on one single center study and it was only published as an abstract (#73, Lion-Francois et al. 2006). A prevalence of 0.25-1% in male ID, as it is mentioned on page 4 in the manuscript, is certainly more realistic.

 We agree with the reviewer that the prevalence of 0.25-1% is more realistic. Thus, we modified the ‘Conclusion and future directions’ paragraph accordingly.

- Literature: The first synopsis of creatine deficiency syndromes has been published in 2003 (Schulze 2003). Although it is based on the first observations in the few patients with AGAT, GAMT and CTD known at the time this synopsis holds true up to the day and could be added to the comprehensive body of literature in this manuscript.

 We apologize for the omission of this important study. We now included this synopsis in the reference list.

Finally, we would like to point out that the other changes that the reviewers will find in the ms are due to the proofreading we made in the attempt to further improve the text quality.

Reviewer 2 Report

Dear Authors,

Thank you for the possibility to review this interesting article. It was a real pleasure to read it carefully. I have no critical remarks.

Author Response

Response to Reviewer #2: (reviewer’s comments are in italics)

Thank you for the possibility to review this interesting article. It was a real pleasure to read it carefully. I have no critical remarks.

 We thank the reviewer for the positive evaluation of our work.

Finally, we would like to point out that the other changes that the reviewers will find in the ms are due to the proofreading we made in the attempt to further improve the text quality.

Reviewer 3 Report

Ghirardini et al. present a comprehensive review on creatine transporter defect, describing the clinical aspects, biochemistry, genetics and treatment. They also reviewed all available data in the literature on animal models. Their literature review is ample on the subject. I would like to ask if they have a reference recommending newborn screening for CTD (line 215) and also in their conclusion, since its treatment is unsatisfactory.

Author Response

Response to Reviewer #3: (reviewer’s comments are in italics)

Ghirardini et al. present a comprehensive review on creatine transporter defect, describing the clinical aspects, biochemistry, genetics and treatment. They also reviewed all available data in the literature on animal models. Their literature review is ample on the subject. I would like to ask if they have a reference recommending newborn screening for CTD (line 215) and also in their conclusion, since its treatment is unsatisfactory.

We thank the reviewer for this comment. However, we do not have a specific reference explicitly recommending newborn screening for CTD. Since misdiagnosis of CTD children has been reported in the literature (e.g., Yildiz et al., 2020, ref. 74), we believe that extending the newborn screening for metabolic disorders to Cr deficiency syndromes would be an important step.

Finally, we would like to point out that the other changes that the reviewers will find in the ms are due to the proofreading we made in the attempt to further improve the text quality.
